



# Reliability Ensemble Averaging of 21st century projections of terrestrial net primary productivity reduces global and regional uncertainties

Jean-François Exbrayat[1], A. Anthony Bloom[2], Pete Falloon[3], Akihiko Ito[4], T. Luke Smallman[1], Mathew Williams[1]

[1]School of GeoSciences and National Centre for Earth Observation, University of Edinburgh, Edinburgh, EH9 3FF, UK
[2]Jet Propulsion Laboratory, California Institute of Technology, Pasadena, California, US
[3]Met Office Hadley Centre, Fitzroy Road, Exeter, EX1 3PB,UK
[4]National Institute for Environmental Studies, Tsukuba, Japan

*Correspondence to*: Jean-François Exbrayat(j.exbrayat@ed.ac.uk)

**Abstract.**Multi-model averaging techniques provide opportunities to extract additional information from large ensembles of simulations. In particular, present-day model skill can be used to evaluate their potential performance in future climate simulations. Multi-model averaging methods have been used extensively in climate and hydrological sciences, but they have not been used to constrain projected plant productivity responses to climate change, which is a major uncertainty in earth system modelling.Here, we use three global observation-orientated estimates of current net primary productivity (NPP) to perform a reliability ensemble averaging (REA) using 30 global simulations of the 21$^{st}$ century change in NPP based on the Inter-Sectoral Impact Model Intercomparison Project (ISI-MIP) 'business as usual' emissions scenario. We find that the three REAs support an increase in global NPP by the end of the 21$^{st}$ century (2090s) compared to the 2000s, which is $4 - 6\%$ stronger than the ensemble ISIMIP mean value of 23.7 Pg C y$^{-1}$. Using REA also leads to a $43 - 67\%$ reduction in the global uncertainty of 21$^{st}$ century NPP projection, which strengthens confidence in the resilience of the $CO_2$-fertilization effect to climate change. This reduction in uncertainty is especially clear for boreal ecosystems. Conversely, the large uncertainty that remains on the sign of the response of NPP in semi-arid regions points to the need for better observations and model development in these regions.

## 1 Introduction

Anthropogenic emissions of carbon dioxide ($CO_2$) enhance the uptake of atmospheric carbon by terrestrial ecosystems through net primary productivity (NPP). This so-called $CO_2$-fertilization effect has helped offset 25-30% of $CO_2$ emissions responsible for climate change in recent decades (Canadell et al., 2007; Le Quéré et al., 2009). There exists a large uncertainty as to whether this positive effect of $CO_2$-fertilization will be resilient to climate change, as shown by the spread between model projections from various intercomparison projects (Friedlingstein et al., 2006; Arora et al., 2013; Friend et al., 2014; Nishina et al., 2014, 2015), especially in highly productive tropical regions (Rammig et al., 2010; Cox et al.,





2013). However, large ensembles of projections are challenging to interpret as they may include models with an opposite response to the same change in boundary conditions (Friedlingstein et al., 2006). Simulations from the Inter-Sectoral Impact Model Intercomparison Project (ISI-MIP, Warszawski et al., 2014) have shown that most of the uncertainty in 21$^{st}$ century projections of the terrestrial carbon cycle can be attributed to differences between global vegetation models (GVMs; Friend et al., 2014; Nishina et al., 2014, 2015), although a non-negligible part of the uncertainty arises from differences in climate projections themselves (Ahlström et al., 2012).

In recent years multi-model averaging has been widely used to extract information from large ensembles of simulations in studies targeting climate change (Bishop and Abramowitz, 2012; Krishnamurti et al., 1999), rainfall-runoff processes (Georgakakos et al., 2004; Huisman et al., 2009; Shamseldin et al., 1997; Viney et al., 2009) and catchment-scale nutrient exports (Exbrayat et al., 2010, 2013b). These methods range from simple arithmetic means of model ensembles to more elaborate weighting schemes that take model performance into account. The underlying assumption is that a model that is better able to reproduce current conditions should be given more weight in the final projection than a poorly performing model. The more complex Reliability Ensemble Averaging (REA; Giorgi and Mearns, 2002) approach takes into account a measure of convergence between projections to identify the most likely change: this way, the REA method avoids giving too much weight to an over-fitted model which may accurately represent current conditions for the wrong reasons but predicts vastly different change than other ensemble members (Exbrayat et al., 2013b). Metrics measuring model independence (Bishop and Abramowitz, 2012) have also been introduced in weighting schemes to avoid pseudo-replication.

Until recently, applying these advanced multi-model averaging methods to simulations of the terrestrial carbon cycle has remained a challenge because of the lack of global observational datasets to constrain e.g. the REA weighting scheme. To our knowledge only Schwalm et al. (2015) have presented results of skill-based model averaging applied to historical simulations of the terrestrial carbon cycle to an ensemble of 10 models from the Multiscale synthesis and Terrestrial Model Intercomparison Project (MsTMIP; Huntzinger et al., 2013). However, we are not aware of any studies using these methods in the context of projections of the terrestrial carbon cycle under climate change.

Here, we present the first example of a spatially-explicit application of the REA approach to extract a best estimate of NPP change (ΔNPP) during the 21$^{st}$ century under a business-as-usual scenario of emissions from a large ensemble of projections. We perform the REA procedure three times using different observation-constrained estimates of current NPP: retrievals of the terrestrial carbon cycle with the CARDAMOM model-data fusion approach (Bloom and Williams, 2015; Bloom et al., 2016), an approximation of NPP based on the up-scaled FLUXCOM GPP datasets (Jung et al., 2009, 2011, 2017; Tramontana et al., 2016), and the MOD17A3 MODIS NPP product (Running et al., 2004; Zhao et al., 2005; Zhao and Running, 2010). Based on optimally-weighted model averages, we evaluate the impact of the REA method on 21$^{st}$ century projections of ΔNPP but also on the uncertainty in the future resilience of the $CO_2$-fertilization that exist among the models. We show that the REA procedure can help identify regions where uncertainties remain large and thereby inform the future development of models and observational networks needed to improve climate change projections.



## 2 Materials & methods

### 2.1 The ISI-MIP ensemble

We used an ensemble of simulations of net primary production (NPP) from the Inter-Sectoral Impact Model Intercomparison Project (ISI-MIP; Warszawski et al., 2014). The ISI-MIP simulations included here consist of 6 global vegetation models:
HYBRID (Friend and White, 2000), JeDi(Pavlick et al., 2013), JULES (Clark et al., 2011), LPJmL (Sitch et al., 2003), SDGVM (Woodward et al., 1995) and VISIT (Ito and Inatomi, 2012). Each of these 6 GVMs was driven by bias-corrected output (Hempel et al., 2013) from 5 general circulation models (GCMs): GFDL-ESM2M (Dunne et al., 2012), HadGEM2-ES (Collins et al., 2011), IPSL-CM5A-LR (Dufresne et al., 2013), MIROC-ESM-CHEM (Watanabe et al., 2011) and NorESM1-M (Bentsen et al., 2013), generating a total of 30 global simulations of NPP for the historical period and under the
representative concentration pathway 8.5 (RCP8.5). We chose the ISI-MIP ensemble over other initiatives like C4MIP (Friedlingstein et al., 2006) or CMIP5 (Taylor et al., 2012) because the combination of multiple GVMs with multiple GCMs in ISI-MIP allows a more comprehensive coverage of the uncertainty in the terrestrial carbon cycle and attribution of dominant factor in the uncertainty of the future (Friend et al., 2014; Nishina et al., 2014, 2015) although we note that these simulations omit feedbacks from the biosphere on weather and atmospheric $CO_2$ concentrations. As the ensemble integrates
5 representations of the same GVM, and 6 representations of the same GCM, we avoid issues related to model genealogy (Knutti et al., 2013) that could lead similar models to bias results of the averaging because of intrinsic lack of independence between the different ensemble members (Bishop and Abramowitz, 2012). We focus our approach on NPP projections under the RCP8.5 scenario of emissions for which more simulations were available (Nishina et al., 2015). Mean annual current NPP and projected changes are summarised in Table 1 and Supplementary Figure S1. We note a large spread in current
global NPP simulated by the models from 51.7 Pg C y$^{-1}$ to 76.5 Pg C y$^{-1}$ during the last 10 years of the historical simulations, as well as ΔNPP in the 2090s ranging from -17.0 to 41.4 Pg C y$^{-1}$. Further information on the models and the ISI-MIP protocol are to be found in the Supplementary Information of Friend et al. (2014) and the respective model description papers listed in Table 1.

### 2.2 Estimates of current NPP

We use three different estimates of current NPP: (a) an observation-bound terrestrial carbon cycle analysis estimate, (b) an estimate based on up-scaled eddy-covariance $CO_2$ flux measurements, and (c) an estimate based on satellite measurements of absorbed photosynthetically active radiation. To harmonize the approach, we re-gridded all observationally-constrained NPP datasets to the lowest dataset resolution (1°×1°), and confined our analysis to the NPP dataset overlap period (2001-2010). Mean annual NPP and variability for each dataset is presented in Figures S2 and S3.



### 2.2.1 CARDAMOM retrievals

The CARbonDAta Model fraMework (Bloom et al., 2016) produces spatially explicit retrievals of the global terrestrial carbon cycle following a model-data fusion procedure. In each 1°×1° pixel, the Data-Assimilation Linked Ecosystem Carbon version 2 (DALEC2; Bloom and Williams, 2015; Williams et al., 2005) is driven by ERA-Interim reanalysis climate data

(Dee et al., 2011) and burned area from the Global Fire Emission Database version 4 (Giglio et al., 2013). A Bayesian Markov Chain Monte-Carlo approach is implemented to constrain DALEC2 according to observations of MODIS leaf area index (Myneni et al., 2002), tropical biomass (Saatchi et al., 2011), soil carbon content from the Harmonized World Soil Database (HWSD; FAO, 2012) and a set of Ecological and Dynamic Constraints (Bloom and Williams, 2015). Through this Bayesian procedure, CARDAMOM provides an explicit estimation of the uncertainty in model parameters and hence in

land-atmosphere carbon fluxes such as net primary production (NPP) from site to global-scale (Bloom et al., 2016; Smallman et al., 2017). However, as not all the other datasets (see sections 2.2.2 and 2.2.3) provide a measure of the parametric uncertainty, in this study we rely on CARDAMOM's highest confidence estimates of a mean annual NPP of 50.1 Pg C y$^{-1}$. More details on the framework can be found in the supplementary information of Bloom et al. (2016).

### 2.2.2 FLUXCOM

The FLUXCOM project uses machine-learning methods (Tramontana et al., 2016) to up-scale global datasets from eddy-covariance measurements of $CO_2$ and energy fluxes from the FLUXNET network (Baldocchi et al., 2001). In a first step, a machine-learning algorithm is used to extract a relationship between local environmental drivers and ecosystem fluxes (Jung et al., 2009). Then, the trained algorithm is used in combination with gridded climate data and remote sensing observations to produce a global estimate of monthly ecosystem fluxes at a 0.5°×0.5° spatial resolution. In its first instance, FLUXCOM

products relied on a random forest method (Breiman, 2001) but newly available datasets have been produced using additional machine learning methods (Tramontana et al., 2016; Jung et al., 2017).

Here, we use the average of an ensemble of six FLUXCOM GPP datasets to derive an estimate of annual NPP for 2001-2010. These datasets were created using three machine-learning methods: random forest, artificial neural networks and multivariate regression splines. Each machine-learning method was used to produce two GPP datasets corresponding to two

partitioning methods of net ecosystem exchange (see Reichstein et al. (2005) and Lasslop et al. (2009)). Then, we used CARDAMOM's retrievals of carbon use efficiency (Bloom et al., 2016), the ratio of NPP to GPP, to derive a current value of NPP of 52.8 Pg C y$^{-1}$ for the first ten years of the 21$^{st}$ century from the 127.1 Pg C y$^{-1}$ FLUXCOM estimated GPP.

### 2.2.3 MODIS NPP

The MOD17 MODIS GPP/NPP dataset provides 8-day estimates of GPP and annual NPP at a 1-km spatial resolution since

the year 2000. Therefore, GPP is calculated as the product of the amount of absorbed photosynthetically active radiation (estimated from the MOD15 MODIS LAI/FPAR product, Myneni et al., 2002) and a biome-specific radiation use efficiency



that is adjusted as a function of air temperature and vapour pressure deficit. Land cover classification is derived from MODIS using the MCD12Q1 product (Friedl et al., 2002) while meteorological data are taken from the National Centers for Environmental Prediction (NCEP)/ Department of Energy (DOE) Reanalyses II. Then, annual maintenance respiration is estimated using a temperature-acclimated $Q_{10}$ relationship (Tjoelker et al., 2001) while growth respiration is assumed to be a

5 fixed fraction of NPP. The MODIS NPP dataset has been used to quantify the impact of droughts (Zhao and Running et al., 2010) and the El Niño/Southern Oscillation on global terrestrial ecosystem productivity (Bastos et al., 2013). We re-gridded the annual NPP data to a 1°×1° spatial resolution for the reference years 2001-2010 from which we derived a 53.6 Pg C $y^{-1}$ mean annual value.

## 2.3 Reliability Ensemble Averaging

Multi-model averaging techniques have been developed to extract information and quantify the uncertainty from large ensembles of simulations (e.g. Krishnamurti et al., 1999). These methods range from simple arithmetic mean to more complex statistical methods (Viney et al., 2009) such as Bayesian Model Averaging (Raftery et al., 2005). A common assumption is that models which better reproduce available observations should be given more weight in a final prediction than poorly performing models. However, models may be over-fitted to match observations, providing the good answers for

the wrong reasons (Exbrayat et al., 2013b). These models are likely to represent improperly, or even omit, processes which may become key under changed conditions, and this challenges their reliability. Therefore, the Reliability Ensemble Averaging method (REA; Giorgi and Mearns, 2002) was developed to integratealso a measure of model convergence in the weighting scheme and penalize models which do not predict the same response to changes (Exbrayat et al., 2013b).

In each 1°×1° pixel, each model projection $i$ of the 30 GVM-GCM ensemble is assigned a reliability factor $R_i$ that is

20 calculated such as

$$R_i = R_{B,i} \times R_{D,i} = \left(\frac{\varepsilon}{|B_i|}\right) \times \left(\frac{\varepsilon}{|D_i|}\right) \qquad (1)$$

where $\varepsilon$ represents the variability in observations expressed as the difference between the largest and smallest values of annual NPP in each pixel (Figure S3; Giorgi and Mearns, 2002), while $B_i$ and $D_i$ correspond to a measure of model $i$'s performance and convergence, respectively. We produce three REA estimates based on CARDAMOM, FLUXCOM and

25 MODIS NPP, further referred to as $REA_C$, $REA_F$ and $REA_M$, respectively. For each REA application, terms $\varepsilon$, $B_i$, $D_i$ and hence $R_i$ (equation 1) are recalculated based on the particular observational dataset to produce three independent sets of model coefficients.

Here, we apply the REA method to the ensemble of 30 ISI-MIP simulations of 21[st] century ΔNPP under RCP8.5 emission scenario. We first re-gridded the ISI-MIP data using the *remapcon* function of the Climate Data Operators version 1.6.9 to

30 match the 1°×1° spatial resolution of the observationally constrained datasets (see section 2.2) and performed the procedure





in each land pixel to create maps of REA averages. We then apply the REA method three times ($REA_C$, $REA_F$ and $REA_M$) to evaluate their current performance.

For each 30 simulations of the ISI-MIP ensemble we calculated $B_i$ in each pixel such as

$$B_i = NPP_i - NPP_{obs} \tag{2}$$

5  where$NPP_i$ is the mean annual NPP predicted by model $i$ during the 10 last years of the historical simulations and $NPP_{obs}$ corresponds to either of the observational datasets mean annual NPP. Then for each model the value of $D_i$ was calculated in each pixel as the difference between the change predicted by model $i$ and the REA average such as

$$D_i = \Delta NPP_i - \frac{\sum_{i=1}^{N} R_i \cdot \Delta NPP_i}{\sum_{i=1}^{N} R_i} \tag{3}$$

where$\Delta NPP_i$ is the change in mean NPP in the last 10 years of the RCP8.5 simulation (2090-2099) compared to the last 10

10  years of the historical simulations (1996-2005) predicted by the ensemble member $i$ and $N$ is the total number of ensemble members. The REA average is not known beforehand and weights $R_{D,i}$ are calculated iteratively(Giorgi and Mearns, 2002). Finally, weights $R_{B,i}$ and $R_{D,i}$ are assigned a maximum value of 1 if the absolute value of $B_i$ and $D_i$ are smaller than $\varepsilon$, the measure of variability in the observations.

The uncertainty around the REA average change is calculated as the weighted root-mean square difference (RMSD)

15  calculated following

$$RMSD = \left( \frac{\sum_{i=1}^{N} R_i \cdot \left( \Delta NPP_i - \Delta NPP_{REA} \right)^2}{\sum_{i=1}^{N} R_i} \right)^{\frac{1}{2}} \tag{4}$$

where$\Delta NPP_{REA}$ is the REA average change. Assuming that the error distribution is somewhere between uniform and Gaussian, the 60-70% confidence interval of the REA is represented by $\Delta NPP_{REA} \pm RMSD$ (Giorgi and Mearns, 2002).

Giorgi and Mearns (2002) further introduced a quantitative measure of the collective model reliability $\rho$, based on $R_i$, where

20  $$\rho = \frac{\sum_{i=1}^{N} R_i^2}{\sum_{i=1}^{N} R_i} \tag{5}$$

which will vary pixel-wise based on each model's performance with respect to the mean and variability represented in each observational dataset as well as the convergence to the REA average. The reliability measure $\rho$ can be further decomposed in $\rho_B$ and $\rho_D$, such as

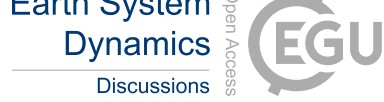



$$\rho_B = \frac{\sum\limits_{i=1}^{N} R_{B,i}}{N} \qquad (6)$$

$$\rho_D = \frac{\sum\limits_{i=1}^{N} R_{D,i}}{N} \qquad (7)$$

where $\rho_B$ and $\rho_D$ correspond to the ensemble reliability with respect to model biases and model convergence respectively.

## 3 Results

The REA averaging method yields a global increase of NPP of 24.7 ± 9.8 Pg C y$^{-1}$(REA average ± RMSD) for CARDAMOM,25.0 ± 10.0 Pg C y$^{-1}$ for FLUXCOM and 23.9 ± 16.2 Pg C y$^{-1}$ for MODIS NPP. As the ISI-MIP ensemble mean indicated a ΔNPP of 23.7 Pg C y$^{-1}$, these results represent a 4% increase of the mean for REA$_C$, 5% for REA$_F$ and 6% for REA$_M$. The pixel-wise one standard deviation uncertainty in the ISI-MIP ensemble was 29.6 Pg C y$^{-1}$ and the REA results indicate strong reduction of 67% for REA$_C$, 66% for REA$_F$ and 43% for REA$_M$. These results further indicate that in

all three cases the REA averaging method reduces the uncertainty of the ensemble spread toward an agreement on a future increase in the global land carbon uptake.

Zonal means (Figure 1) indicate that the ISI-MIP ensemble mean and all three REA$_C$, REA$_F$ and REA$_M$ averages estimate an increase in NPP across all latitudes. All three REA averages predict a weaker increase in NPP at high latitudes of the northern and southern hemispheres. They also agree on a stronger increase in NPP than the ISI-MIP ensemble mean for

tropical regions between 15°S and 10°N but also between 20°N and 25°N and temperate regions around 45°N. REA$_C$ indicates a weaker increase in NPP than ISI-MIP around 20°S while REA$_F$ and REA$_M$ averages are similar to the ISI-MIP ensemble mean in these regions. The uncertainty around each of the REA averages is smaller than the uncertainty around the ISI-MIP ensemble mean across all latitudinal zones. Furthermore, while the very large uncertainty around the ISI-MIP ensemble mean does not provide confidence on the sign of ΔNPP across most regions, the uncertainty around all three REA

averages is constrained toward an increase in NPP across all regions, except around 20°S.

The spatial distribution of the ISI-MIP ensemble mean ΔNPP contrasts with that of the three REA averages with noticeable differences across all regions of the globe (Figure 2). All three REA averages predict a weaker increase in NPP than the ISI-MIP ensemble in Canada and Scandinavia, while they predict a stronger increase in NPP in Eurasia. Similarly, all three REA averages predict a stronger increase in NPP than the ISI-MIP ensemble in tropical rainforest of South America, Africa and

south-east Asia. The REA averages agree on a weaker ΔNPP in semi-arid regions of the Sahel, southern Africa, Australia and the Tibetan Plateau. Overall, all REA$_C$, REA$_F$ and REA$_M$ exhibit broadly similar patterns in the spatial distribution of ΔNPP differences with the ISI-MIP ensemble mean that is confirmed by R$^2$ values of 0.74 between REA$_C$ and REA$_F$, 0.66 between REA$_C$ and REA$_M$ and 0.70 between REA$_F$ and REA$_M$.



The uncertainty in ΔNPP is reduced across most regions of the globe for all three $REA_C$, $REA_F$ and $REA_M$ (Figure 1 and Figure 3). This reduction of uncertainty leads to a confidence on the sign estimation of ΔNPP in 84%, 80% and 73% of all the land pixels for $REA_C$, $REA_F$ and $REA_M$ respectively, against 35% for the ISI-MIP ensemble. The average reduction in uncertainty is large in regions north of 40°N (Figure 1), mostly corresponding to a reduction in uncertainty in boreal Eurasia (Figure 3) that provides better confidence in an increase in NPP (Figure 2). We note that the uncertainty in the $REA_M$ remains similar to the uncertainty around the ISI-MIP ensemble mean for large portions of the southern hemisphere such as southern Africa. However, all three $REA_C$, $REA_F$ and $REA_M$ cannot provide confidence on the sign of ΔNPP for southern Africa and Australia.

The zonal means of the mean values of the three coefficients $R_i$, $R_{B,i}$ and $R_{D,i}$ (Figure 4) show that MODIS-based $REA_M$ yields larger values of all coefficients compared to $REA_C$ and $REA_F$. We note strong inter-model similarities in the spatial distribution of model weights ($R_i$; Figure 4a-c), biases ($R_{B,i}$; Figure 4d-f) and convergence of the projected ΔNPP ($R_{D,i}$; Figure 4g-i). Only the HYBRID models are almost systematically assigned lower weight $R_i$ as a result of lower values for both $R_{B,i}$ (i.e. a larger bias than the other models) and $R_{D,i}$ (i.e. a divergence in projected ΔNPP). This is especially obvious in boreal regions north of 60°N where HYBRID is assigned values close to 0 in $REA_C$ and $REA_F$.

The collective model reliability measure $\rho$ provides a quantification of the spread of model weights determined through the REA method (Figure 5). Regions where $\rho$ is close to 1 indicates places where there is a strong consensus between models on the current NPP but also on 21$^{st}$ century ΔNPP. There are large differences in $\rho$ depending on the NPP observational datasets using to constrain the REA (Figure 5). Indeed, while the average value of $\rho$ is 0.35 for $REA_C$ and 0.38 for $REA_F$, it is 0.75 for $REA_M$. $REA_C$ and $REA_F$ yields very low values of $\rho$ in boreal regions (Figure 5) while $REA_M$ leads to values of $\rho$ close to 1 in most regions south of 60°S. The measure of reliability $\rho$ can be further decomposed in two components $\rho_B$ and $\rho_D$ (Figure 5d-i, equations 6 and 7). Results indicate that $\rho_D$ is consistently greater than $\rho_B$ for all $REA_C$, $REA_F$ and $REA_M$. This result means that model convergence in the simulation of ΔNPP is greater than the model performance to reproduce current NPP. In other words, the model performance evaluated against the three current NPP datasets contributes the most to decreasing the ensemble reliability $\rho$. Values of $\rho_B$ are lower than 0.10 in boreal regions for $REA_C$ and $REA_F$, indicating that model bias is greater than the variability of NPP $\varepsilon$ estimated from the CARDAMOM retrievals and the FLUXCOM based NPP by a factor 10. Conversely, regions where $\rho_B$ is close to 1 for $REA_M$ indicate that the variability in the MODIS NPP observations is larger than model biases.

## 4 Discussion

The globally integrated values of the REA average change (23.9 to 25.0 Pg C y-1) and the ISI-MIP ensemble mean (23.7 Pg C y-1) are similar. This is in agreement with a previous multi-model approach that only found a 0.01 Pg C y-1 difference in historical mean annual net ecosystem exchange between a simple mean and a weighted average based on model performance (Schwalm et al., 2015). However, by contrast with this previous study, we find that in all three $REA_C$, $REA_F$ and $REA_M$ a



large spatial variability in grid cell differences (Figure 2) that compensate each other to yield a relatively small global difference with ISI-MIP ensemble mean. The three REA averages indicate a stronger positive ΔNPP than the ISI-MIP ensemble mean for boreal Eurasia and tropical rainforests (Figures 1 and 2), and a weaker but still positive ΔNPP in northern Canada and semi-arid regions like the Sahel, the Tibetan plateau, southern Africa and Australia.

The reduction in uncertainty arising from the REA method helps putting a greater confidence in a sustained $CO_2$-fertilization effect throughout the 21st century although these results may be influenced by model-wise differences in process representation. In both the ISI-MIP ensemble mean and the three REA averages, the sustained increase of NPP at high latitudes, where nitrogen (N) limitation on NPP dominates (Zhang et al., 2011; Exbrayat et al., 2013a) but is only represented in the HYBRID and SDGVM models (Table 1; Nishina et al., 2014). The increase in NPP in these N-limited regions is in

contrast with observations at Free-Air $CO_2$ Enrichment experiments that indicate a quick weakening of the $CO_2$-fertilization effect as soil N stores deplete (Norby et al., 2010). Models which integrate coupled C-N cycles generally predict the historical land carbon sink in good agreement with estimates from the Global Carbon Budget (Huntzinger et al., 2017) and project a decrease in NPP throughout the 21st century (Thornton et al., 2009; Goll et al., 2012; Zhang et al., 2013; Wieder et al., 2015).

Similarly, recent observations have concluded a total absence of $CO_2$-fertilization effect under phosphorus-limited conditions (Ellsworth et al., 2017) which dominates in the tropics and leads to an additional reduction of NPP in model projections (Goll et al., 2012; Zhang et al., 2013; Wieder et al., 2015). Here, only the HYBRID and SDGVM models integrate the representation of N limitations on NPP (Nishina et al., 2014) and none of them represent phosphorous limitations. HYBRID is also the only model to predict a possible decrease in global NPP throughout the 21st century (Table

1 and Friend et al., 2014) because of a reduction at high latitudes and in tropical rainforests (Supplementary Figure S1). Thus, HYBRID is assigned low $R_{D,i}$ weights in these regions (Figure 4g-i and Supplementary Figures S4-12) and cannot influence the REA average and the calculation of its uncertainty (equation 4) despite integrating more detailed representation of ecosystem processes. However, HYBRID also exhibits stronger differences to the observational datasets than the other models especially at high latitudes (Figure 4d-f) which may indicate a strong sensitivity of N limitations. Nevertheless, we

note that all models' performances tend to decrease in regions north of 60°N where their ΔNPP projections also diverge (Figure 4g-i, Figure 5g-i). Overall, we note that the promising REA results should be used carefully as they cannot correct for the omissions of key processes by a large fraction of the ensemble members. There is also considerable debate on how good large-scale NPP observational products are (Kolby-Smith et al., 2015; de Kauwe et al., 2016), a problem that we address by performing the REA approach three times.

In all three $REA_C$, $REA_F$ and $REA_M$ cases, the global uncertainty around the REA average is reduced compared to the uncertainty within the ISI-MIP ensemble which provides a higher degree of confidence in the resilience of the global $CO_2$-fertilization effect to warming. The reduction in uncertainty, and the gain in confidence on the sign of ΔNPP, is especially obvious in boreal regions for all three REA (Figure 3). Conversely, uncertainties on the sign of ΔNPP remain large for all REA in semi-arid regions of Southern Africa and Australia. It is a non-trivial result as the response of these ecosystems to





climate events like El Niño and La Niña drives the inter-annual variability and the trend of the global terrestrial carbon sink (Bastos et al., 2013; Poulter et al., 2014; Ahlström et al., 2015), while projections indicate a gain of forest ecosystems over savannahs in the future (Moncrieff et al., 2016).

Because of the way the REA method assigns coefficients to ensemble members with respect to the annual variability in the data $\varepsilon$ (equation 1), the final REA average and uncertainty are conditional on the variability represented in current estimate of NPP. Figure 5a-c shows that the reliability of the ensemble measured by $\rho$ varies depending on which observational dataset is used, although generally lower values of $\rho B$ and $\rho D$ at high latitudes indicate that models disagree on the current NPP and future $\Delta$NPP in these regions. Furthermore, high values of $\rho$ for REAM indicate a larger variability $\varepsilon$ in the MODIS dataset compare to CARDAMOM and the FLUXCOM based NPP data (Figure S3). This larger variability leads to more models being given a weight close to 1 in the averaging scheme because the variability is larger than their bias (Figure 5f) or the predicted change (Figure 5i). Conversely, the relatively smaller variability in CARDAMOM retrievals leads more models to be weighted poorly according to both their performance (Figure 5d) and their convergence with other models (Figure 5g). The variability $\varepsilon$ influences the final uncertainty and as a result the REA$_C$ has a smaller uncertainty because it is more penalizing on models, and vice-versa with MODIS NPP.

## 5 Conclusion

We applied the REA method on a pixel-by-pixel base to an ensemble of 30 simulations of historical and 21st century NPP from the ISI-MIP project. Our results indicate that using either CARDAMOM retrievals, a FLUXCOM based estimate of current NPP or data from MODIS to constrain the REA scheme helps at least halving the uncertainty in 21st century global $\Delta$NPP. This process leads to a higher confidence in a sustained CO2-fertilization effect. We nevertheless note that a large uncertainty remains in semi-arid regions that is mostly attributable to differences in process representation in global vegetation models. Furthermore, most models used here do not account for N limitations on NPP and this may have altered the outcome of the convergence coefficient used in REA.

**Acknowledgements**

This work was supported by the Natural Environment Research Council through the National Centre for Earth Observation. Part of this work was carried out at the Jet Propulsion Laboratory, California Institute of Technology, under a contract with the National Aeronautics and Space Administration. PF was supported by the Joint UK DECC/Defra Met Office Hadley Centre Climate Programme (GA01101). For their roles in producing, coordinating, and making available the ISI-MIP model output, we acknowledge the modelling groups and the ISI-MIP coordination team.



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





**Tables**

**Table 1: Information about global vegetation models used here. For each GVMs we indicate the range of values obtained while driving it with 5 GCMs.**

| Model | NPP (1996-2005) Pg C y$^{-1}$ | $\Delta$NPP Pg C y$^{-1}$ | Nitrogen[a] | Reference |
|---|---|---|---|---|
| HYBRID | 63.5 – 76.1 | -17.0 – 25.3 | Yes | Friend and White (2000) |
| JeDi | 55.5 – 63.8 | 24.6 – 32.3 | No | Pavlick et al. (2013) |
| JULES | 65.1 – 71.5 | 34.1 – 41.4 | No | Clark et al. (2011) |
| LPJ | 69.6 – 75.6 | 26.7 – 35.0 | No | Sitch et al. (2003) |
| SDGVM | 70.9 – 74.8 | 32.3 – 37.5 | Yes | (Woodward et al., 1995) |
| VISIT | 51.7 – 59.7 | 29.1 – 32.3 | No | Ito and Inatomi (2012) |

[a]from Nishina et al. (2015)



**Figures**

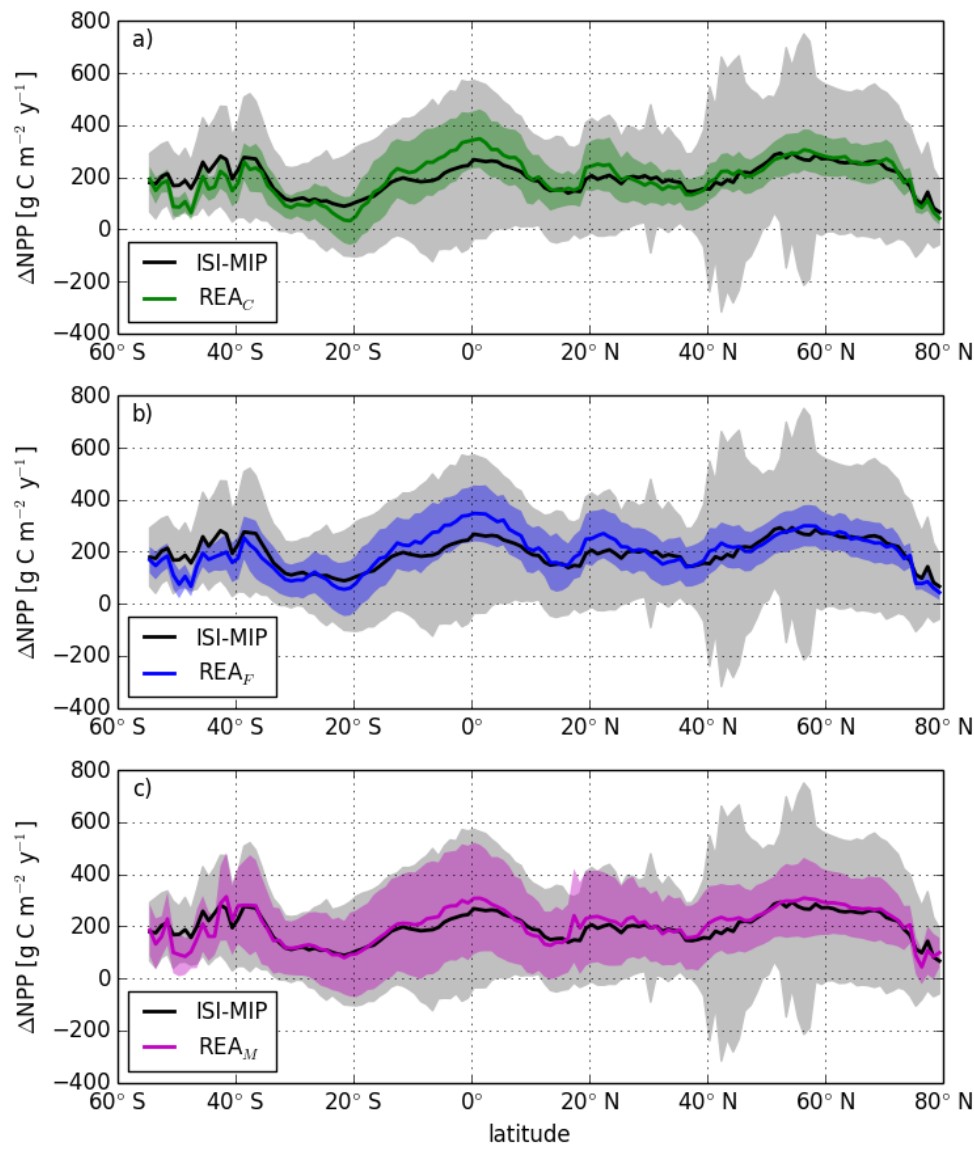

**Figure 1: Zonal mean ΔNPP by the end of the 21st century under RCP8.5 compared to historical simulations. Shading represents the uncertainty around the zonal mean across the ISI-MIP ensemble, taken as one standard deviation for ISI-MIP, and calculated following equation (4) for REA. REA$_C$, REA$_F$ and REA$_M$, refer to REA values calculated based on observationally-constrained CARDAMOM, FLUXCOM and MODIS NPP respectively.**



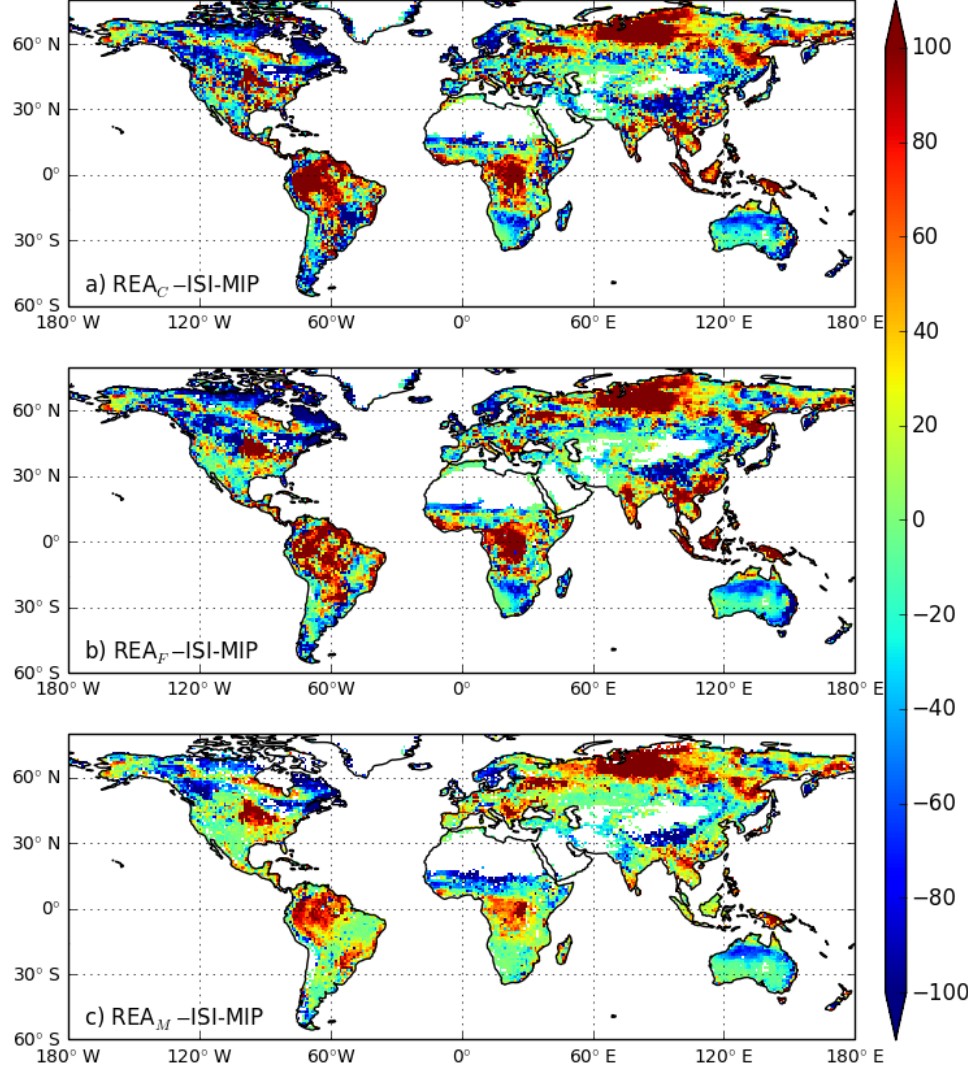

**Figure 2: Differences between ΔNPP from REA average and ISI-MIP ensemble mean (in g C m$^{-2}$ y$^{-1}$). Red indicates where the REA averages predict ΔNPP greater than the ISI-MIP ensemble mean. Blue indicates where the REA averages predict ΔNPP less than the ISI-MIP ensemble mean.**





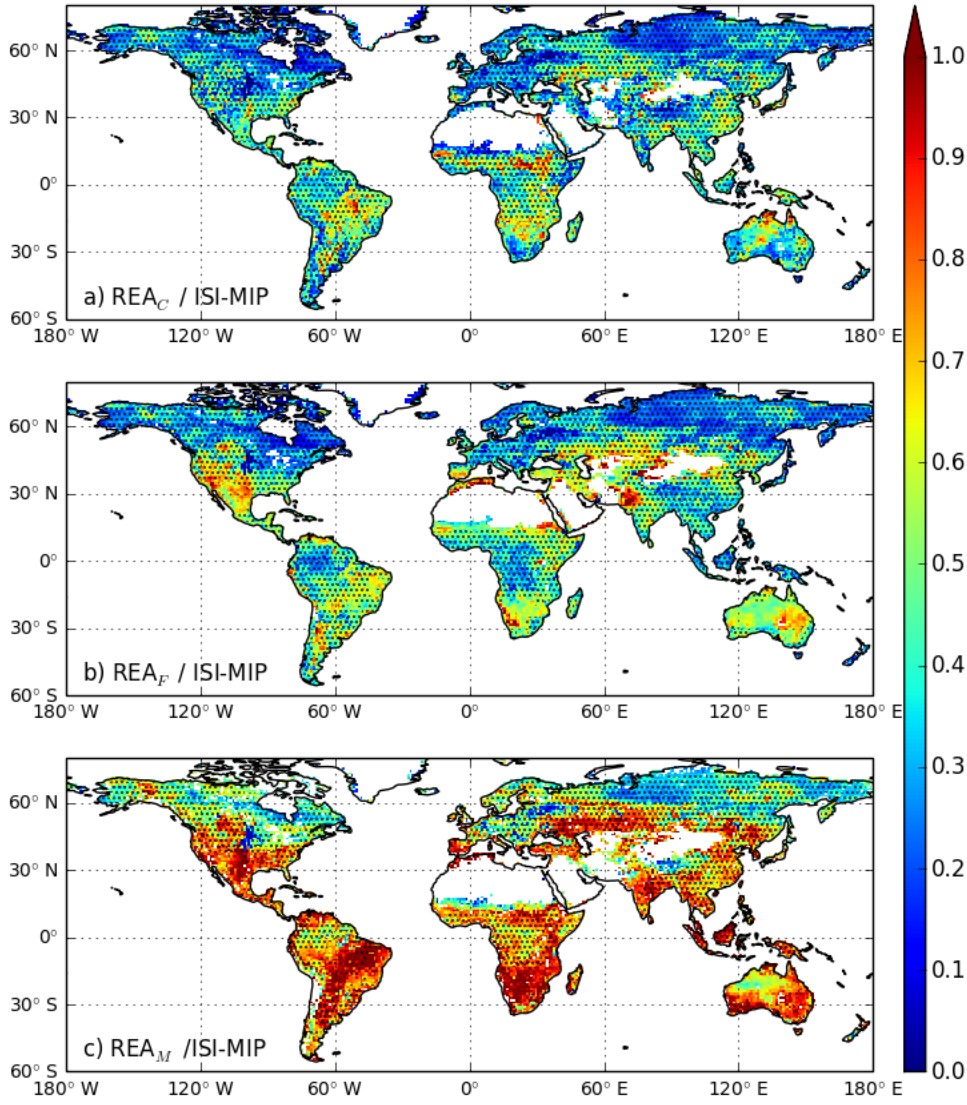

**Figure 3:** Ratio of the uncertainty from each REA to the uncertainty in the ISI-MIP ensemble. For ISI-MIP, the uncertainty is calculated as the standard deviation across the ensemble while the uncertainty around the REA averages is calculated following equation 4. Stippling indicates regions where there is an agreement on the sign of ΔNPP through the uncertainty.



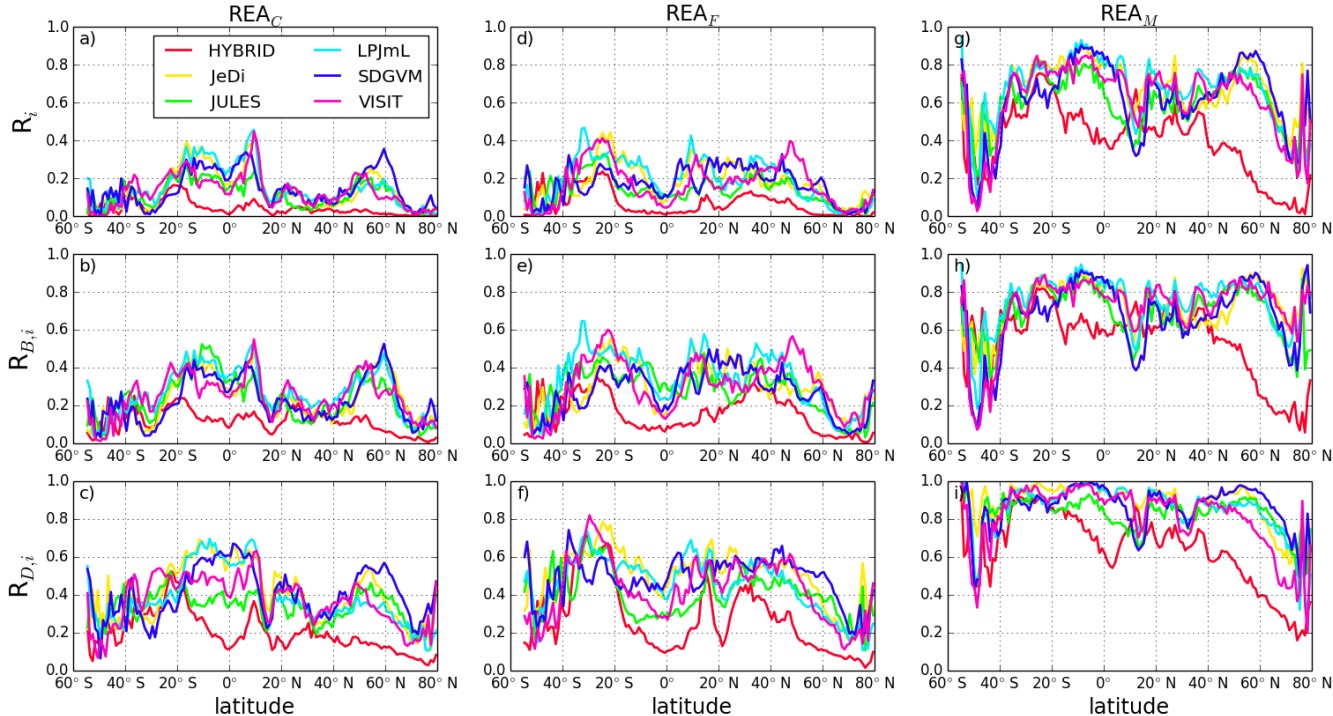

**Figure 4: Zonal mean $R_i$, $R_{B,i}$ and $R_{D,i}$ (row-wise) in each REA$_C$, REA$_F$ and REA$_M$ (column-wise). Each line represents the average value obtained across the five simulations of each GVM.**





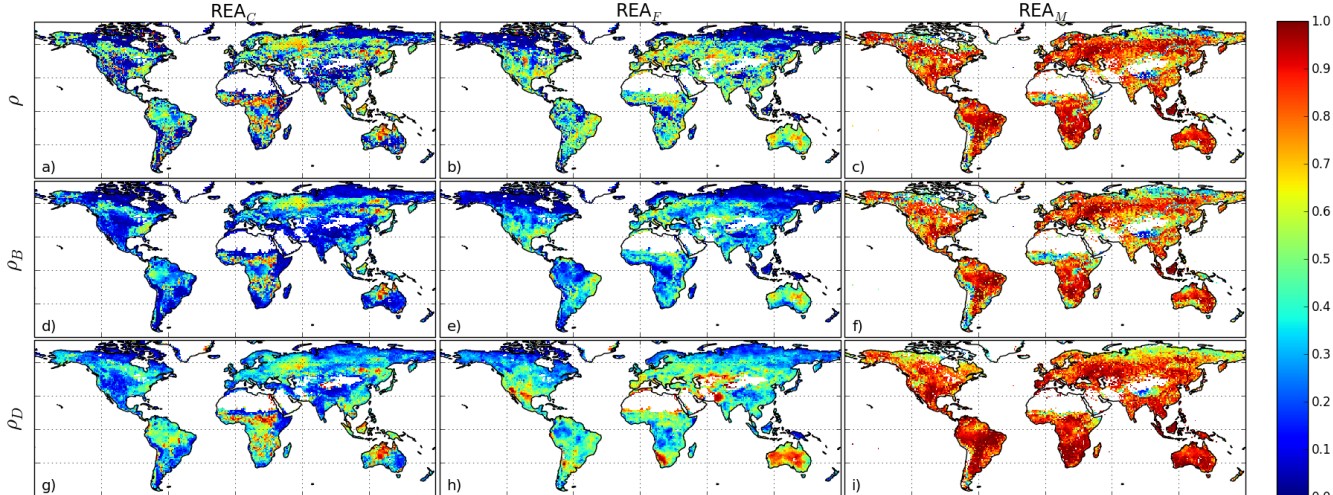

**Figure 5: Collective model reliability ρ, model performance ρB and model convergence ρD (row-wise) for each REA$_C$, REA$_F$ and REA$_M$ (column-wise).**