# Peer review of "Reliability Ensemble Averaging of 21st century projections of terrestrial net primary productivity reduces global and regional uncertainties"

_Earth System Dynamics, 2017_

## Referee Comment (RC1) · Anonymous Referee #1 · 17 Oct 2017

This paper describes statistical analysis of ISIMIP NPP dataset which weights models by their present day/historical performance in order to constrain the range of future estimates of NPP change. This is a well written and generally very clear disposition. I have a couple of small queries about the text, but no major issues.

Comments:

Given the current popularity of the emergent constraint methodology, it would be useful to have a brief compare/contrast of how this method differs, as they seem superficially

similar.

The paper does an excellent job of explaining in appropriate detail the methods, but on page 6, line 1 three REAs are listed, but not explained what they are. It becomes clear in a figure caption later, but it would be good to explain in here too.

I'd like to see a nod towards the uncertainties of the analysis in the abstract, particularly the lack of key processes (nitrogen, phosphorus, etc.) in the DGVMs. The discussion is good on this, but the abstract portrays a more uncritical acceptance of the reduction of uncertainty in the high latitudes, (especially boreal systems), which isn't completely supported by the data.

A brief discussion of the limits of this technique - especially regards whether we're increasing the precision but not the accuracy of the projections – would be useful. This is especially important given the issue about process representation, and the low weighting of the HYBRID model.

The map colour schemes are eye wateringly terrible, as well as not being colour blind friendly. The green in the middle makes it really difficult to read the plots accurately. The figure 4 plots would be enhanced by using different line patterns as well as colour, to help people read it in black and white print as well as colour blind readers. A cursory google or ask around the office should get the authors decent colour schemes. It's really not acceptable to use rainbow anymore.

There's a slightly higher than average number of words without spaces between them. This just needs checking.

---

## Author Comment (AC1) · 3 Nov 2017

This paper describes statistical analysis of ISIMIP NPP dataset which weights models by their present day/historical performance in order to constrain the range of future estimates of NPP change. This is a well written and generally very clear disposition. I have a couple of small queries about the text, but no major issues.

Dear Reviewer

Many thanks for the review and the helpful comments on the approach on the manuscript. In the following, we provide an initial answer to your comments and will include additional text in the revised manuscript.

Comments:
Given the current popularity of the emergent constraint methodology, it would be useful to have a brief compare/contrast of how this method differs, as they seem superficially similar.

We agree with the reviewer that some aspects of the Reliability Ensemble Averaging (REA) are similar to multi-model averaging methods previously used in the context of terrestrial carbon cycle (e.g. Schwalm et al., 2015; Lovenduski and Bonan, 2017). Indeed, like in these recent studies, REA assigns more weight to simulations made by models that are more skilled to reproduce past observations. However, REA also considers how projections compare to each other by providing a measure of the convergence around the weighted average.

Beyond differences in the weighting schemes themselves, we also note discrepancies in the type and number of constraints, resolution and time period considered among studies. Lovenduski and Bonan (2017) consider a single value of cumulative terrestrial carbon uptake for 1959-2005 to derive one global coefficient per model. We apply the REA scheme on a pixel-by-pixel base using three different estimates of the same process while Schwalm et al. (2015) use multiple constraints on stocks and fluxes in each land pixel. However, we consider $21^{st}$ century projections using a pixel-wise approach while Schwalm et al. (2015) focus on historical simulations.

We will review these aspects in relevant parts of the introduction, methods and discussion of the revised manuscript.

The paper does an excellent job of explaining in appropriate detail the methods, but on page 6, line 1 three REAs are listed, but not explained what they are. It becomes clear in a figure caption later, but it would be good to explain in here too.

We refer to $REA_C$, $REA_F$ and $REA_M$ as the three REA cases driven by CARDAMOM, FLUXCOM and MODIS, respectively already on p. 5 l. 24-25. However, we take this comment as a necessity to remind the reader of the definition of each of the $REA_C$, $REA_F$ and $REA_M$

throughout the text for improved clarity and we will do so in a revised version of the manuscript.

I'd like to see a nod towards the uncertainties of the analysis in the abstract, particularly the lack of key processes (nitrogen, phosphorus, etc.) in the DGVMs. The discussion is good on this, but the abstract portrays a more uncritical acceptance of the reduction of uncertainty in the high latitudes, (especially boreal systems), which isn't completely supported by the data.

We agree that this is one of the major findings/limitations of our approach and needs to be highlighted in the abstract. We will add the following sentence to the abstract:

> This reduction in uncertainty is especially clear for boreal ecosystems although it may be an artefact due to the lack of representation of nutrient limitations on NPP in most models.

A brief discussion of the limits of this technique - especially regards whether we're increasing the precision but not the accuracy of the projections – would be useful. This is especially important given the issue about process representation, and the low weighting of the HYBRID model.

We agree that the low $R_{D,i}$ assigned to HYBRID at low and high latitudes may be due to its explicit representation of nitrogen limitations on NPP. These leads HYBRID to be the only model to project a possible decrease in global NPP by the end of the century and it becomes an outlier that is penalised by low values of $R_{D,i}$ (p.9 l. 20-23).

Overall, the outcome of the REA approach cannot account for missing processes and remains conditional on the ensemble to which it is applied. This involves a risk to increase the precision around some inaccurate projections if treated like a black box. Following this comment, the revised manuscript will emphasise that REA outcomes should be cautiously interpreted with respect to the ensemble members.

The map colour schemes are eye wateringly terrible, as well as not being colour blind friendly. The green in the middle makes it really difficult to read the plots accurately. The figure 4 plots would be enhanced by using different line patterns as well as colour, to help people read it in black and white print as well as colour blind readers. A cursory google or ask around the office should get the authors decent colour schemes. It's really not acceptable to use rainbow anymore.

We take this comment very seriously. Therefore, we provide new Figures 2 to 5 using a colour scheme that is compatible with colour-blindness (checked on http://www.vischeck.com ) and renders well on black and white printers. We attach these updated figures at the end of this document and will correct the Supplementary Information accordingly upon submission of a revised manuscript.

There's a slightly higher than average number of words without spaces between them. This just needs checking.

We believe that this is an issue with the conversion of the original document into a pdf. We will double check upon submission of the revised manuscript.

References

Lovenduski, N. S. and Bonan, G. B.: Reducing uncertainty in projections of terrestrial carbon uptake, Env. Res. Lett., 12(4), 044020, 2017.

Schwalm, C. R., Huntzinger, D. N., Fisher, J. B., Michalak, A. M., Bowman, K., Ciais, P., Cook, R., El-Masri, B., Hayes, D., Huang, M., Ito, A., Jain, A., King, A. W., Lei, H., Liu, J., Lu, C., Mao, J., Peng, S., Poulter, B., Ricciuto, D., Schaefer, K., Shi, X., Tao, B., Tian, H., Wang, W., Wei, Y., Yang, J. and Zeng, N.: Toward "optimal" integration of terrestrial biosphere models, Geophys. Res. Lett., 42(11), 4418–4428, doi:10.1002/2015GL064002, 2015.

[Figure]

Figure 2: Differences between ΔNPP from REA average and ISI-MIP ensemble mean (in g C m$^{-2}$ y$^{-1}$). Red indicates where the REA averages predict ΔNPP greater than the ISI-MIP ensemble mean. Blue indicates where the REA averages predict ΔNPP less than the ISI-MIP ensemble mean.

[Figure]

Figure 3: Ratio of the uncertainty from each REA to the uncertainty in the ISI-MIP ensemble. For ISI-MIP, the uncertainty is calculated as the standard deviation across the ensemble while the uncertainty around the REA averages is calculated following equation 4. Stippling indicates regions where there is an agreement on the sign of ΔNPP through the uncertainty.

[Figure]

Figure 4: Zonal mean $R_i$, $R_{B,i}$ and $R_{D,i}$ (row-wise) in each $REA_C$, $REA_F$ and $REA_M$ (column-wise). Each line represents the average value obtained across the five simulations of each GVM.

[Figure]

Figure 5: Collective model reliability ρ, model performance $\rho_B$ and model convergence $\rho_D$ (row-wise) for each $REA_C$, $REA_F$ and $REA_M$ (column-wise).

---

## Referee Comment (RC2) · Anonymous Referee #2 · 7 Jan 2018

The paper talks about a different approach of reliability ensemble averaging to calculate the average of multi-model estimates of global NPP for future scenario RCP 8.5. This new methodology takes into consideration 2 important aspects while allocating weights to different model estimates for calculating the ensemble mean: performance of the models as compared to the observations and convergence measure. Overall, introducing a new approach to calculate ensemble mean from different model estimates on a global scale is commendable and significant at this point in time when the world is focussing on quantifying the carbon fluxes for future and uncertainties in these estimates are large posing a challenge for scientists to come up with ways of reducing them. The analysis of the results obtained is extensive and comprehensive. However, there are some concerns that seem to be important.

Specific Comments:

In the discussion section, the major point that has been highlighted is the lack of representation of other elements, specifically N, in the GVMs used in this study and how their availability can limit carbon sequestration by vegetation in future. This has also been supported by multiple studies cited in the text. From the point of view of scientific knowledge and the focus on reduction in uncertainty from model estimates, the fact that of the 6 GVMs used in this study, only 2 (HYBRID and SDGVM) include the impact of N on model NPP estimation does not give a lot of reliability on results of this study. There should be some possible explanation for this difference in results of this study (increase in NPP) from other studies (reduction in NPP due to N limitation) to make the results more acceptable and reliable. In terms of introducing a new method for computing averages, the study has done a good job, but in terms of reliability and accuracy of the results of this study, it is questionable. This is a major concern.

There are different time periods that are included in the text. For instance, data from the 3 datasets used (CARDAMOM, FLUXCOM, MODIS) are from 2001-2010. While calculating $B_i$ in equation (2), the difference between model predictions during last 10 years of historical simulations (1996-2005) and NPP from observations (2001-2010) is considered, or so it seems. It would be good to clarify why 2 different time periods are considered for calculating the performance measure ($B_i$) of models with observed values. Ideally, a comparison should be done for the same time period.

Captions of figures should be improved to include details like time period for which the given figure represents mean. For instance, in the caption of figure 1, what years comprise the historical simulation can be added. Captions should be as complete in themselves as possible.

Title of section 2.2 on page 3 'Estimates of current NPP' is confusing since the ISI-MIP model simulations also include the current period.

In the manuscript, appropriate spaces have been missed between 2 words or a word and a full stop. Like in page 5 line 17, the word 'integratealso'. The authors are advised to go through the text and revise these typographical mistakes.

In section 2.3 on Reliability Ensemble Averaging, before the actual method has been described there is a lot of description of the other methods used for calculating mean. This part from line 10 to 16 on page 5 can be a part of the introduction, where it identifies why these other methods are not serving the purpose and there is a need for a better strategy. Since REA is the method finally adopted in this study, the description of only this method used should be a part of this section 2.3.

Since REA is a new approach introduced for calculating NPP in this study, it would be good if the terms in equation (1) and (5) are described in terms of their maximum and minimum possible values, and their significance to give a more meaningful perspective of this approach.

---

## Author Comment (AC2) · 12 Jan 2018

The paper talks about a different approach of reliability ensemble averaging to calcu- late the average of multi-model estimates of global NPP for future scenario RCP 8.5. This new methodology takes into consideration 2 important aspects while allocating weights to different model estimates for calculating the ensemble mean: performance of the models as compared to the observations and convergence measure. Overall, introducing a new approach to calculate ensemble mean from different model estimates on a global scale is commendable and significant at this point in time when the world is focussing on quantifying the carbon fluxes for future and uncertainties in these estimates are large posing a challenge for scientists to come up with ways of reducing them. The analysis of the results obtained is extensive and comprehensive. However, there are some concerns that seem to be important.

Dear Reviewer,

Thank you for your insightful comments that will help improve the manuscript. We provide an initial answer to your comments in the following, and will include some additional text in a revised version of the manuscript.

Specific Comments:
In the discussion section, the major point that has been highlighted is the lack of representation of other elements, specifically N, in the GVMs used in this study and how their availability can limit carbon sequestration by vegetation in future. This has also been supported by multiple studies cited in the text. From the point of view of scientific knowledge and the focus on reduction in uncertainty from model estimates, the fact that of the 6 GVMs used in this study, only 2 (HYBRID and SDGVM) include the impact of N on model NPP estimation does not give a lot of reliability on results of this study. There should be some possible explanation for this difference in results of this study (increase in NPP) from other studies (reduction in NPP due to N limitation) to make the results more acceptable and reliable. In terms of introducing a new method for computing averages, the study has done a good job, but in terms of reliability and accuracy of the results of this study, it is questionable. This is a major concern.

Multi-model averaging is a post-processing procedure aiming at extracting knowledge from existing large ensemble of simulations. Like in previous multi-model averaging studies focused on the carbon cycle (e.g. Schwalm et al., 2015; Lovenduski and Bonan, 2017) or climate (Krishnamurti et al., 1999; Giorgi and Mearns, 2002) we used already available simulations in a "post-MIP" exercise. Overall, the outcome of the REA approach cannot account for missing processes and remains conditional on the ensemble to which it is applied. It is therefore beyond the scope of this paper to resolve the lack of process representation in some GVMs.

Nevertheless, we agree that the lack of representation of nutrient limitations on NPP in 4 out of 6 GVMs used here is a concern considering the possible implications for future productivity in response to increase $CO_2$ concentrations (e.g. Wieder et al., 2015), a point we had already made in the discussion. We note, however, that this 1/3 ratio of models including carbon-nutrient interactions in the ISI-MIP ensemble is commensurate to other MIPs: 3 out of 12 CMIP5 models used by Todd-Brown et al. (2014), 2 out of 8 models in new ISI-MIP experiments presented by Chen et al. (2017). Furthermore, low weights $R_i$ assigned to HYBRID (Figure 4a-c), which includes

carbon-nutrient interactions, are not only due to a lack of convergence with the other models (Figure 4g-i) but also because of its poorer agreement with observational datasets (Figure 4d-f). SDGVM, the other model that includes carbon-nutrient interactions, is more similar to the carbon-only models in terms of historical performance and projected changes.

Overall, we accept this comment as a need to better explain the origin of the simulations and the post-processing nature of the averaging approach in the revised manuscript.

There are different time periods that are included in the text. For instance, data from the 3 datasets used (CARDAMOM, FLUXCOM, MODIS) are from 2001-2010. While calculating Bi in equation (2), the difference between model predictions during last 10 years of historical simulations (1996-2005) and NPP from observations (2001-2010) is considered, or so it seems. It would be good to clarify why 2 different time periods are considered for calculating the performance measure (Bi) of models with observed values. Ideally, a comparison should be done for the same time period.

We agree that the benchmarking period should be the same. Therefore, we have redone the experiments using the time period 2001-2005 to evaluate Bi. As a result, we now compare the 2001-2005 reference period to the last five years of the projections for 2095-2099. Results are similar and numbers will be updated throughout the manuscript. For example, the first paragraph of the results section will now read (updated numbers in red):

The REA averaging method yields a global increase of NPP of 24.6 ± 8.5 Pg C $y^{-1}$ (REA average ± RMSD) for CARDAMOM, 24.8 ± 9.5 Pg C $y^{-1}$ for FLUXCOM and 25.0 ± 14.5 Pg C $y^{-1}$ for MODIS NPP. As the ISI-MIP ensemble mean indicated a ΔNPP of 24.2 Pg C $y^{-1}$, these results represent a ~2% increase of the mean for both $REA_C$ and $REA_F$ and 3% for $REA_M$. The pixel-wise one standard deviation uncertainty in the ISI-MIP ensemble was 26.3 Pg C $y^{-1}$ and the REA results indicate strong reduction of 68% for $REA_C$, 64% for $REA_F$ and 45% for $REA_M$. These results further indicate that in all three cases the REA averaging method reduces the uncertainty of the ensemble spread toward an agreement on a future increase in the global land carbon uptake.

Captions of figures should be improved to include details like time period for which the given figure represents mean. For instance, in the caption of figure 1, what years comprise the historical simulation can be added. Captions should be as complete in themselves as possible.

We will improve figure captions to include more detailed descriptions. For example, the caption of figure 1 will now read (updated text in red):

Figure 1: Zonal mean ΔNPP by the end of the 21$^{st}$ century (averaged over 2095 to 2099) under RCP8.5 compared to the end of the historical simulations (averaged over 2001 to 2005). Shading represents the uncertainty around the zonal mean across the ISI-MIP ensemble, taken as one standard deviation for ISI-MIP, and calculated following equation (4) for REA. $REA_C$, $REA_F$ and $REA_M$, refer to REA values calculated based on observationally-constrained CARDAMOM, FLUXCOM and MODIS NPP respectively.

Title of section 2.2 on page 3 'Estimates of current NPP' is confusing since the ISI-MIP model simulations also include the current period.

We will replace with "Benchmark datasets of modern NPP".

In the manuscript, appropriate spaces have been missed between 2 words or a word and a full stop. Like in page 5 line 17, the word 'integratealso'. The authors are advised to go through the text and revise these typographical mistakes.

We note that this comment is similar to reviewer #1's and will make sure that these typos will disappear in the revised manuscript.

In section 2.3 on Reliability Ensemble Averaging, before the actual method has been described there is a lot of description of the other methods used for calculating mean. This part from line 10 to 16 on page 5 can be a part of the introduction, where it identifies why these other methods are not serving the purpose and there is a need for a better strategy. Since REA is the method finally adopted in this study, the description of only this method used should be a part of this section 2.3.

We agree that this section of the text is misplaced, and actually redundant with the text page l. 7 to 17. Therefore, we will remove it from the method section.

Since REA is a new approach introduced for calculating NPP in this study, it would be good if the terms in equation (1) and (5) are described in terms of their maximum and minimum possible values, and their significance to give a more meaningful perspective of this approach.

Terms $R_i$, $R_{B,i}$ and $R_{D,i}$ are model weights and range from 0, for a poorly performing model, to 1. As noted p 6 l 12-13:

> Finally, weights $R_{B,i}$ and $R_{D,i}$ are assigned a maximum value of 1 if the absolute value of $B_i$ and $D_i$ are smaller than $\varepsilon$, the measure of variability in the observations.

We will move the above closer to equation 1 and will include a better description of the range in the revised manuscript.

**References**

Chen, M., Rafique, R., Asrar, G. R., Bond-Lamberty, B., Ciais, P., Zhao, F., Reyer, C. P. O., Ostberg, S., Chang, J., Ito, A., Yang, J., Zeng, N., Kalnay, E., West, T., Leng, G., Francois, L., Munhoven, G., Henrot, A., Tian, H., Pan, S., Nishina, K., Viovy, N., Morfopoulos, C., Betts, R., Schaphoff, S., Steinkamp, J. and Hickler, T.: Regional contribution to variability and trends of global gross primary productivity, Environ. Res. Lett., 12(10), doi:10.1088/1748-9326/aa8978, 2017.

Giorgi, F. and Mearns, L. O.: Calculation of Average, Uncertainty Range, and Reliability of Regional Climate Changes from AOGCM Simulations via the "Reliability Ensemble Averaging" (REA) Method, J. Clim., 15, 1141–1158, doi:10.1175/1520-0442(2002)015<1141:COAURA>2.0.CO;2, 2002.

Krishnamurti, T. N., Kishtawal, C. M., LaRow, T. E., Bachiochi, D. R., Zhang, Z., Williford, C. E., Gadgil, S. and Surendran, S.: Improved Weather and Seasonal Climate Forecasts from Multimodel Superensemble, Science (80-. )., 285(5433), 1548–1550, doi:10.1126/science.285.5433.1548, 1999.

Lovenduski, N. S. and Bonan, G. B.: Reducing uncertainty in projections of terrestrial carbon uptake, Env. Res. Lett., 12(4), 044020, 2017.

Schwalm, C. R., Huntzinger, D. N., Fisher, J. B., Michalak, A. M., Bowman, K., Ciais, P., Cook, R., El-Masri, B., Hayes, D., Huang, M., Ito, A., Jain, A., King, A. W., Lei, H., Liu, J., Lu, C., Mao, J., Peng, S., Poulter, B., Ricciuto, D., Schaefer, K., Shi, X., Tao, B., Tian, H., Wang, W., Wei, Y., Yang, J. and Zeng, N.: Toward "optimal" integration of terrestrial biosphere models, Geophys. Res. Lett., 42(11), 4418–4428, doi:10.1002/2015GL064002, 2015.

Todd-Brown, K. E. O., Randerson, J. T., Hopkins, F., Arora, V., Hajima, T., Jones, C., Shevliakova, E., Tjiputra, J., Volodin, E., Wu, T., Zhang, Q. and Allison, S. D.: Changes in soil organic carbon storage predicted by Earth system models during the 21st century, Biogeosciences, 11(8), 2341–2356, doi:10.5194/bg-11-2341-2014, 2014.

Wieder, W. R., Cleveland, C. C., Smith, W. K. and Todd-Brown, K.: Future productivity and carbon storage limited by terrestrial nutrient availability, Nat. Geosci., 8(6), 441–444, doi:10.1038/NGEO2413, 2015.

---

## Author Response (AR1)

Please find below our replies to both reviewers followed by a revised manuscript with tracked changes.

Reviewer #1

This paper describes statistical analysis of ISIMIP NPP dataset which weights models by their present day/historical performance in order to constrain the range of future estimates of NPP change. This is a well written and generally very clear disposition. I have a couple of small queries about the text, but no major issues.

Dear Reviewer

Many thanks for the review and the helpful comments on the approach on the manuscript. In the following, we provide an initial answer to your comments and will include additional text in the revised manuscript.

Comments:
Given the current popularity of the emergent constraint methodology, it would be useful to have a brief compare/contrast of how this method differs, as they seem superficially similar.

We agree with the reviewer that some aspects of the Reliability Ensemble Averaging (REA) are similar to multi-model averaging methods previously used in the context of terrestrial carbon cycle (e.g. Schwalm et al., 2015; Lovenduski and Bonan, 2017). Indeed, like in these recent studies, REA assigns more weight to simulations made by models that are more skilled to reproduce past observations. However, REA also considers how projections compare to each other by providing a measure of the convergence around the weighted average.

Beyond differences in the weighting schemes themselves, we also note discrepancies in the type and number of constraints, resolution and time period considered among studies. Lovenduski and Bonan (2017) consider a single value of cumulative terrestrial carbon uptake for 1959-2005 to derive one global coefficient per model. We apply the REA scheme on a pixel-by-pixel base using three different estimates of the same process while Schwalm et al. (2015) use multiple constraints on stocks and fluxes in each land pixel. However, we consider 21$^{st}$ century projections using a pixel-wise approach while Schwalm et al. (2015) focus on historical simulations.

We have added these aspects p2 l. 18-26:

> Until recently, applying these advanced multi-model averaging methods to simulations of the global carbon cycle has remained a challenge because of the lack of global observational datasets to constrain e.g. the REA weighting scheme. Schwalm et al. (2015) have presented results of skill-based model averaging applied to an ensemble of 10 models from the Multiscale synthesis and Terrestrial Model Intercomparison Project (MsTMIP; Huntzinger et al., 2013). This pixel-wise approach assigned weights to historical simulations based on their performance to simulate gross primary productivity and biomass stocks but did not consider future projections. Lovenduski and Bonan (2017) considered a single value of cumulative terrestrial carbon uptake for 1959-2005 to

derive one global coefficient per model to produce new projections. However, we are not aware of any studies using these methods in the context of spatially-explicit projections of the terrestrial carbon cycle under climate change.

The paper does an excellent job of explaining in appropriate detail the methods, but on page 6, line 1 three REAs are listed, but not explained what they are. It becomes clear in a figure caption later, but it would be good to explain in here too.

We refer to $REA_C$, $REA_F$ and $REA_M$ as the three REA cases driven by CARDAMOM, FLUXCOM and MODIS, respectively already on p. 5 l. 26-27. However, we take this comment as a necessity to remind the reader of the definition of each of the $REA_C$, $REA_F$ and $REA_M$ throughout the text and we do so p7 l. 7-8:

> The REA averaging method yields a global increase of NPP of 24.6 ± 8.5 Pg C y-1 (REA average ± RMSD) using CARDAMOM in $REA_C$, 24.8 ± 9.5 Pg C y-1 using FLUXCOM in $REA_F$ and 25.0 ± 14.4 Pg C y-1 using MODIS NPP in $REA_M$.

I'd like to see a nod towards the uncertainties of the analysis in the abstract, particularly the lack of key processes (nitrogen, phosphorus, etc.) in the DGVMs. The discussion is good on this, but the abstract portrays a more uncritical acceptance of the reduction of uncertainty in the high latitudes, (especially boreal systems), which isn't completely supported by the data.

We agree that this is one of the major findings/limitations of our approach and needs to be highlighted in the abstract. We have added the following sentence to the abstract p1 l 21-22:

> This reduction in uncertainty is especially clear for boreal ecosystems although it may be an artefact due to the lack of representation of nutrient limitations on NPP in most models.

A brief discussion of the limits of this technique - especially regards whether we're increasing the precision but not the accuracy of the projections – would be useful. This is especially important given the issue about process representation, and the low weighting of the HYBRID model.

We agree that the low $R_{D,i}$ assigned to HYBRID at low and high latitudes may be due to its explicit representation of nitrogen limitations on NPP. These leads HYBRID to be the only model to project a possible decrease in global NPP by the end of the century and it becomes an outlier that is penalised by low values of $R_{D,i}$. However, HYBRID also performs less well than the other models as shown by low values of $R_{B,i}$ on Figure 4. Both these aspects play in the overall low $R_i$ assigned to HYBRID as noted in the discussion p. 9 l 18-25:

> HYBRID is also the only model to predict a possible decrease in global NPP throughout the 21st century (Table 1 and Friend et al., 2014) because of a reduction at high latitudes and in tropical rainforests (Supplementary Figure S1). Thus, HYBRID is assigned low $R_{D,i}$ weights in these regions (Figure 4g-i and Supplementary Figures S4-12) and cannot influence the REA average and the calculation of its uncertainty (equation 4) despite integrating more detailed representation of ecosystem processes. However, HYBRID also exhibits stronger differences to the observational

datasets than the other models especially at high latitudes (Figure 4d-f) which may indicate a strong sensitivity of N limitations. Nevertheless, we note that all models' performances tend to decrease in regions north of 60°N where their $\Delta$NPP projections also diverge (Figure 4d-f, Figure 5d-f)

Overall, the outcome of the REA approach cannot account for missing processes and remains conditional on the ensemble to which it is applied. This involves a risk to increase the precision around some inaccurate projections if treated like a black box. Following this comment and a similar comment from reviewer #2, we have added the following to the discussion p. 9 l 26-31:

Overall, the promising REA results should be used carefully as they cannot correct for the omission of key processes by a large fraction of the ensemble members. Like in previous multi-model averaging studies focused on the carbon cycle (e.g. Schwalm et al., 2015; Lovenduski and Bonan, 2017) or climate (Krishnamurti et al., 1999; Giorgi and Mearns, 2002) we used already available simulations in a post-processing procedure. We note, however, that the ratio of two out of six models including carbon-nutrient interactions in the ISI-MIP ensemble is commensurate to other model inter-comparison projects: 3 out of 10 CMIP5 models (Exbrayat et al., 2014) or 2 out of 8 models in the new ISI-MIP experiments presented by Chen et al. (2017).

The map colour schemes are eye wateringly terrible, as well as not being colour blind friendly. The green in the middle makes it really difficult to read the plots accurately. The figure 4 plots would be enhanced by using different line patterns as well as colour, to help people read it in black and white print as well as colour blind readers. A cursory google or ask around the office should get the authors decent colour schemes. It's really not acceptable to use rainbow anymore.

We take this comment very seriously. Therefore, we have replaced Figures 2 to 5 using colour schemes that are compatible with colour-blindness (checked on http://www.vischeck.com) and have updated Figure 4 with line patterns. We have corrected the Supplementary Information accordingly.

There's a slightly higher than average number of words without spaces between them. This just needs checking.

We believe that this is an issue with the conversion of the original document into a pdf. We will double check upon submission of the revised manuscript.

Reviewer #2

The paper talks about a different approach of reliability ensemble averaging to calcu- late the average of multi-model estimates of global NPP for future scenario RCP 8.5. This new methodology takes into consideration 2 important aspects while allocating weights to different model estimates for calculating the ensemble mean: performance of the models as compared to the observations and convergence measure. Overall, introducing a new approach to calculate ensemble mean from different model estimates on a global scale is commendable and significant at this point in time when the world is focussing on quantifying the carbon fluxes for future and uncertainties in these estimates are large posing a challenge for scientists to come up with ways of reducing them. The analysis of the results obtained is extensive and comprehensive. However, there are some concerns that seem to be important.

Dear Reviewer,

Thank you for your insightful comments that will help improve the manuscript. We provide an initial answer to your comments in the following, and will include some additional text in a revised version of the manuscript.

Specific Comments:
In the discussion section, the major point that has been highlighted is the lack of representation of other elements, specifically N, in the GVMs used in this study and how their availability can limit carbon sequestration by vegetation in future. This has also been supported by multiple studies cited in the text. From the point of view of scientific knowledge and the focus on reduction in uncertainty from model estimates, the fact that of the 6 GVMs used in this study, only 2 (HYBRID and SDGVM) include the impact of N on model NPP estimation does not give a lot of reliability on results of this study. There should be some possible explanation for this difference in results of this study (increase in NPP) from other studies (reduction in NPP due to N limitation) to make the results more acceptable and reliable. In terms of introducing a new method for computing averages, the study has done a good job, but in terms of reliability and accuracy of the results of this study, it is questionable. This is a major concern.

Multi-model averaging is a post-processing procedure aiming at extracting knowledge from existing large ensemble of simulations. Like in previous multi-model averaging studies focused on the carbon cycle (e.g. Schwalm et al., 2015; Lovenduski and Bonan, 2017) or climate (Krishnamurti et al., 1999; Giorgi and Mearns, 2002) we used already available simulations in a "post-MIP" exercise. Overall, the outcome of the REA approach cannot account for missing processes and remains conditional on the ensemble to which it is applied. It is therefore beyond the scope of this paper to resolve the lack of process representation in some GVMs.

Nevertheless, we agree that the lack of representation of nutrient limitations on NPP in 4 out of 6 GVMs used here is a concern considering the possible implications for future productivity in response to increase $CO_2$ concentrations (e.g. Wieder et al., 2015), a point we had already made in the discussion. We note, however, that this 1/3 ratio of models including carbon-nutrient interactions in the ISI-MIP ensemble is commensurate to other MIPs: 2 out of 10 CMIP5 models used by Exbrayat et al. (2014), 2 out of 8 models

in new ISI-MIP experiments presented by Chen et al. (2017). Furthermore, low weights Ri assigned to HYBRID (Figure 4a-c), which includes carbon-nutrient interactions, are not only due to a lack of convergence with the other models (Figure 4g-i) but also because of its poorer agreement with observational datasets (Figure 4d-f). SDGVM, the other model that includes carbon-nutrient interactions, is more similar to the carbon-only models in terms of historical performance and projected changes.

Overall, we accept this comment as a need to better explain the origin of the simulations and the post-processing nature of the averaging approach and we do so in the revised discussion p.9 l 26-31:

> Overall, the promising REA results should be used carefully as they cannot correct for the omission of key processes by a large fraction of the ensemble members. Like in previous multi-model averaging studies focused on the carbon cycle (e.g. Schwalm et al., 2015; Lovenduski and Bonan, 2017) or climate (Krishnamurti et al., 1999; Giorgi and Mearns, 2002) we used already available simulations in a post-processing procedure. We note, however, that the ratio of two out of six models including carbon-nutrient interactions in the ISI-MIP ensemble is commensurate to other model inter-comparison projects: 3 out of 10 CMIP5 models (Exbrayat et al., 2014) or 2 out of 8 models in the new ISI-MIP experiments presented by Chen et al. (2017).

There are different time periods that are included in the text. For instance, data from the 3 datasets used (CARDAMOM, FLUXCOM, MODIS) are from 2001-2010. While calculating $B_i$ in equation (2), the difference between model predictions during last 10 years of historical simulations (1996-2005) and NPP from observations (2001-2010) is considered, or so it seems. It would be good to clarify why 2 different time periods are considered for calculating the performance measure ($B_i$) of models with observed values. Ideally, a comparison should be done for the same time period.

We agree that the benchmarking period should be the same. Therefore, we have redone the experiments using the time period 2001-2005 to evaluate $B_i$. As a result, we now compare the 2001-2005 reference period to the last five years of the projections for 2095-2099. Results are similar and numbers have been updated throughout the manuscript. For example, the first paragraph of the results section now reads p.7 l. 7-13:

> The REA averaging method yields a global increase of NPP of 24.6 ± 8.5 Pg C y-1 (REA average ± RMSD) using CARDAMOM in REAC, 24.8 ± 9.5 Pg C y-1 using FLUXCOM in REAF and 25.0 ± 14.4 Pg C y-1 using MODIS NPP in REAM. As the ISI-MIP ensemble mean indicated a ΔNPP of 24.2 Pg C y-1, these results represent a ~2% increase of the mean for both REAC and REAF and 3% for REAM. The pixel-wise one standard deviation uncertainty in the ISI-MIP ensemble was 26.3 Pg C y-1 and the REA results indicate strong reduction of 68% for REAC, 64% for REAF and 45% for REAM. These results further indicate that in all three cases the REA averaging method reduces the uncertainty of the ensemble spread toward an agreement on a future increase in the global land carbon uptake.

Captions of figures should be improved to include details like time period for which the given figure represents mean. For instance, in the caption of figure 1, what years comprise the historical simulation can be added. Captions should be as complete in themselves as possible.

We have improved the figure captions to include more detailed descriptions. For example, the caption of figure 1 now reads (p. 19):

> Figure 1: Zonal mean ΔNPP by the end of the 21st century (averaged over 2095-2099) under RCP8.5 compared to the end of the historical simulations (averaged over 2001-2005). Shading represents the uncertainty around the zonal mean across the ISI-MIP ensemble, taken as one standard deviation for ISI-MIP, and calculated following equation (4) for REA. REAC, REAF and REAM, refer to REA values calculated based on observationally-constrained CARDAMOM, FLUXCOM and MODIS NPP respectively.

Title of section 2.2 on page 3 'Estimates of current NPP' is confusing since the ISI-MIP model simulations also include the current period.

We have replaced with "Benchmark datasets of modern NPP" (p. 3 l. 27)

In the manuscript, appropriate spaces have been missed between 2 words or a word and a full stop. Like in page 5 line 17, the word 'integratealso'. The authors are advised to go through the text and revise these typographical mistakes.

We note that this comment is similar to reviewer #1's and will make sure that these typos will disappear in the revised manuscript.

In section 2.3 on Reliability Ensemble Averaging, before the actual method has been described there is a lot of description of the other methods used for calculating mean. This part from line 10 to 16 on page 5 can be a part of the introduction, where it identifies why these other methods are not serving the purpose and there is a need for a better strategy. Since REA is the method finally adopted in this study, the description of only this method used should be a part of this section 2.3.

We agree that this section of the text is misplaced, and actually redundant with the text page 2 l. 7 to 18. Therefore, we will remove it from the method section that now starts with the following (p5 l. 14-17):

> The Reliability Ensemble Averaging method (REA; Giorgi and Mearns, 2002) was developed to assign coefficients to models in the context of future projections. Additionally to using a measure of model performance to reproduce historical conditions, the REA weighting scheme implements measure of model convergence to penalize models that do not predict the same response to changes (Exbrayat et al., 2013b).

Since REA is a new approach introduced for calculating NPP in this study, it would be good if the terms in equation (1) and (5) are described in terms of their maximum and minimum possible values, and their significance to give a more meaningful perspective of this approach.

Terms $R_i$, $R_{B,i}$ and $R_{D,i}$ are model weights and range from 0, for a poorly performing model, to 1. We have improved the explanation of the possible range taken by these three terms p.5 l 23-26:

> The performance coefficient $R_{B,i}$ ranges from 0, for a poorly performing model, to 1 if the absolute value of $B_i$ is smaller than the variability $\varepsilon$. Similarly, the convergence coefficient $R_{D,i}$

ranges from 0 for outlier projections to 1 if the absolute value of $D_i$, the difference between the projection and the REA mean, is smaller than $\varepsilon$. As a result, the final model weight $R_i$ also takes values ranging from 0 to 1.

[revised manuscript text omitted]